# Quality and Utility of Information Captured by Surveillance Systems Relevant to Antimicrobial Resistance (AMR): A Systematic Review

**DOI:** 10.3390/antibiotics10040431

**Published:** 2021-04-13

**Authors:** Mustafa Al-Haboubi, Rebecca E. Glover, Elizabeth Eastmure, Mark Petticrew, Nick Black, Nicholas Mays

**Affiliations:** 1Department of Health Services Research and Policy, London School of Hygiene and Tropical Medicine, London WC1H 9SH, UK; rebecca.glover@lshtm.ac.uk (R.E.G.); elizabeth.eastmure@lshtm.ac.uk (E.E.); nick.black@lshtm.ac.uk (N.B.); nicholas.mays@lshtm.ac.uk (N.M.); 2Department of Public Health, Environments and Society, London School of Hygiene and Tropical Medicine, London WC1E 7HT, UK; mark.petticrew@lshtm.ac.uk

**Keywords:** antimicrobial resistance, surveillance systems, attributes

## Abstract

Health surveillance systems are considered vital for combatting antimicrobial resistance (AMR); however, the evidence-base on the effectiveness of these systems in providing information that can be used by healthcare professionals, or the acceptability of these systems by users, has not been reviewed. A systematic review was conducted of a number of databases to synthesise the evidence. The review identified 43 studies that met the inclusion criteria, conducted in 18 countries and used 11 attributes in their assessment of surveillance systems. The majority of systems evaluated were for monitoring the incidence of tuberculosis. The studies found that most surveillance systems were underperforming in key attributes that relate to both effectiveness and acceptability. We identified that two features of systems (ease of use and users’ awareness of systems) were associated with greater acceptability and completeness of systems. We recommend prioritising these for the improvement of existing systems, as well as ensuring consistency in the definition of attributes studied, to allow a more consistent approach in evaluations of surveillance systems, and to facilitate the identification of the attributes that have the greatest impact on the utility of data produced.

## 1. Introduction

The process of antimicrobial resistance (AMR), whereby microbes evolve over time to become less susceptible to medicines, has reduced the ability of healthcare providers to treat common infections. It is a naturally occurring process that takes place through genetic mutations. The main drivers for AMR include the misuse and overuse of antimicrobials; lack of access to clean water, sanitation and hygiene; poor infection and disease prevention and control; poor access to quality, affordable medicines, vaccines and diagnostics; and lack of enforcement of legislation [1]. In the context of combating AMR, the World Health Organisation (WHO) views surveillance systems as being essential for providing information on the magnitude and trends in AMR and for monitoring the effects of interventions [2].

The most widely recognised guidelines for establishing the utility of the outputs of a surveillance system are those developed by the US Centers for Disease Control and Prevention (CDC) [3]. These describe the usefulness of a system in terms of its contribution to the prevention and control of adverse health events, including an understanding of the implications of those events, based on consideration of nine attributes: Simplicity, Flexibility, Data Quality, Acceptability, Sensitivity, Positive Predictive Value (PPV), Representativeness, Timeliness, and Stability (See Appendix A for definitions). The overall usefulness of a surveillance system is considered to be influenced (to varying degrees depending on the aim of the system) by these attributes [3]. For example, a public health surveillance system that is simple, flexible, acceptable, and stable is more likely to be useful for public health action. However, the CDC acknowledges that there is no perfect system and that focusing resources to improve one attribute might have an adverse effect on another; for example, as sensitivity increases, the PPV might decrease, and efforts to increase sensitivity and PPV could result in a more complex surveillance system with reduced acceptability and timeliness [3]. There is uncertainty about what constitute the most effective characteristics of a system.

A rapid review of the literature identified no systematic reviews of evaluations of the ability of surveillance systems to provide information that can be used by health care professionals to combat AMR. Such a synthesis could contribute to the evidence base when deciding whether to invest or continue investing in these systems in the future. Hence, the aim of this systematic review was to synthesise the evidence from evaluations of the quality and utility of information produced by surveillance systems that monitor organisms and conditions important in the AMR context, with a focus on human rather than animal surveillance systems. The research questions were:What is the effectiveness of AMR-relevant surveillance systems in providing information that can be used to inform healthcare professionals?What is the acceptability of these systems to users?

## 2. Materials and Methods

The study protocol was registered with the International Prospective Register of Systematic Reviews (PROSPERO), registration number: CRD42018085346.

In order to answer the first research question (the effectiveness of systems), the following study designs were eligible for inclusion:Prospective observational studies (controlled and uncontrolled before and after studies).Retrospective observational evaluations, including case-control studies, retrospective cohort studies, and audits. Data sources included primary data collected for research and secondary data (for example, health insurance claim data).Interventions using an experimental design.

Evidence from qualitative research (such as interviews and focus groups) was considered eligible for answering the second research question on the acceptability of systems.

Evaluations of surveillance systems that monitored the following organisms were eligible for inclusion:Bacteria whose antibiotic susceptibility status was recorded by the surveillance system.Bacteria relevant to AMR. A list was collated from the key AMR threats that have been identified by the WHO [4], the CDC [5], European Centre for Disease Control (ECDC) [6], European Food Safety Authority (EFSA) [7], and the key drug-bug combinations identified by Public Health England in the UK AMR Strategy [8].The following types of evaluation study were excluded from the review:Evaluations of public surveillance systems that monitor non-bacterial microorganisms (for example, viruses or fungi).Evaluations of surveillance systems that monitor bacterial microorganisms that are not on any of the priority lists described in the inclusion criteria above.Screening systems that are limited to a single or group of hospitals, and where the information is not shared outside the hospital system.Studies published prior to 1988, when the first CDC guidelines for evaluating Public Health Surveillance systems were published.Articles published in languages other than English.

The following databases were searched for relevant articles from 1 January 1988 until 1 June 2018: OVID Medline; EMBASE; Cochrane Central Register of Controlled Trials (CENTRAL); Global Health (OVID); Web of Science; Open Grey and Scopus.

The search terms used can be found in Appendix B. They were adapted for databases in accordance with the repository’s interface and search options. All search strings were run in English, and all the records were exported to Endnote reference management software v 18.0.2 and Excel 2016. In addition to searching databases, we also performed reference searches of the identified.

### 2.1. Outcomes

The primary outcome for inclusion in the review, to answer the research question on effectiveness, were attributes of surveillance systems, as described by the authors. Aspects of acceptability, such as ease of use, reported in qualitative data, were the outcomes of interest for answering the second research question.

### 2.2. Quality Assessment of Studies

The Critical Appraisal Skills Programme (CASP) checklist [9] was used for assessing the quality of reporting of the qualitative studies included in the review, and the Effective Public Health Practice Project (EPHPP) Quality Assessment Tool was used for those that included a quantitative component [10].

### 2.3. Analysis

Given the heterogeneity among study designs and contexts, a narrative review was conducted following guidelines for narrative synthesis [11].

## 3. Results

### 3.1. Outcome of Study Identification Process

A total of 49,000 records were identified from databases and manual searches, of which 43 studies were included in the review. Figure 1 provides the details of the records excluded at each stage of the screening process.

### 3.2. Characteristics of Included Studies

#### 3.2.1. Study Design

The majority of studies were exclusively quantitative (38/43), with the other five including a qualitative component. None of the studies was purely qualitative. The majority were retrospective analyses of routinely collected data (41/43) (Table 1). Of these, five combined retrospective quantitative analysis with semi-structured interviews and another five incorporated capture/recapture methods to measure the completeness of data.

In over half of the studies (29/43), the system under investigation was compared to other sources of data (laboratory reports, prescription data, medical records, or health insurance claims data) or alternative notification technologies. These comparators had been treated by researchers in two different ways:Use of another surveillance system (such as the CDC Emerging Infections Programme) as a high-quality reference standard against which to compare (for example, Nguyen et al. [12])Comparison between different methods of data collection and reporting, including comparing electronic reporting against other forms of reporting, for example, Saeed et al. [13].

The sample size of the included studies, which in most instances was the number of patients or samples, ranged from 35 in an evaluation of a system for reporting active Tuberculosis (TB) among US military service members [14] to a maximum of 251,693 records in an evaluation of TB surveillance in Brazilian micro-regions [15].

#### 3.2.2. Quality of Studies

The CASP checklist [9] of qualitative research identified a number of problems in the five studies that included such methods [13,15,16,17,18]. These included the suitability of using a qualitative approach for analysing predominately closed questions [13,15] and the absence in all studies of details of how data were collected or analysed. However, all studies included an introductory section that stated clearly the aims of the study.

The quality assessment of the studies that included a quantitative component, using the Effective Public Health Practice Project (EPHPP) Quality Assessment Tool for Quantitative Studies [19], resulted in all of the studies receiving an overall rating of weak. This was primarily the result of the non-randomised sampling cases from systems and the lack of clarity over how representative the samples (case records and/or time periods covered) were of the wider system under investigation.

An analysis of the potential for sampling bias in the 30 studies [12,18,20,21,22,23,24,25,26,27,28,29,30,31,32,33,34,35,36,37,38,39,40,41,42,43,44,45,46,47] that used retrospective analysis of routinely collected data revealed that the rationale for the choice of the time period and the geographic area was not provided in all studies. Furthermore, the studies did not all include a description of who had collected and analysed the data and whether they had an affiliation with the programme (i.e., whether they were independent or not). These are potential sources of data collection and analysis bias. However, studies that included the use of the capture-recapture statistical method to measure the completeness of systems [14,42,48,49,50] all acknowledged the potential sources of bias that could result in the violation of any of the principles that underlie the capture-recapture method [51].

#### 3.2.3. Setting

The evaluations included took place in 18 countries (Table 2), with the largest number in the US (ten studies), followed by Australia and the UK (four evaluations each). The included articles were published between 1991 and 2016.

#### 3.2.4. Surveillance Systems’ Attributes Evaluated

The 43 studies included used 18 attributes in their assessment of health surveillance systems. However, on inspecting the description of these attributes and how they were applied in the studies, they could be grouped into 11, as some authors were using different terms to describe the same attributes. For example, the term “completeness” was sometimes used to describe the proportion of cases of a condition that were detected by the surveillance system under examination. The same definition was used to describe “sensitivity” in other studies. Furthermore, the authors used the same term to describe different characteristics of systems. For example, “completeness” (defined above) was also used to describe the extent to which forms containing the details of each case identified were complete. A description of attributes as they were used in the studies included in this review can be found in Table 3.

The number of attributes examined in each evaluation ranged from one to ten. The attribute most commonly used (20/43) was “completeness” (the proportion of cases identified by the system). Appendix C provides the list of attributes used in each evaluation.

Two of the attributes identified in the systematic review were not identified in the CDC guidelines [3]; these were concordance and specificity. However, specificity was described by the CDC as influencing the positive predictive value (an attribute included in the CDC list).

#### 3.2.5. Health Conditions and Microorganisms Monitored by Systems

The majority of surveillance systems were monitoring the incidence of TB (22/43). Table 4 provides details of the number of evaluations by health condition. Appendix D lists the attributes examined by health condition or microorganism.

### 3.3. Performance of Systems in Relation to Attributes Assessed

The findings of the narrative synthesis are presented for each of the 11 attributes of surveillance systems (as defined in Table 3) examined in relevant studies, starting with the attributes that relate to effectiveness.

#### 3.3.1. Specificity

The only study to consider this attribute [39] was an evaluation of the National Mycobacterial Surveillance System that screened for TB among new refugees in New South Wales, Australia. The study used retrospectively collected data and reported that in nearly one-third of the cases notified as having active TB (*n* = 60), the individuals did not actually have active disease, which suggests low specificity and over-estimation of disease levels.

#### 3.3.2. Usefulness

Saeed et al. [13] was the only included study to look at this attribute. It investigated the possible use of the data generated by two surveillance systems in Afghanistan: the Health Management Information System (HMIS) and the National Tuberculosis Control Programme (NTP), using interviews with key informants. They reported that the HMIS data were useful for planning and monitoring but less so for detecting outbreaks (as a result of poor coordination with the national programme). In contrast, proper analysis of the data from the NTP system could detect and allow a response to outbreaks, as they are available to stakeholders at both national and international levels.

#### 3.3.3. Completeness

As identified above, the term “completeness” was used to describe two distinct attributes: the proportion of cases of the condition that are picked up by the system (also referred to as sensitivity or coverage); and the extent to which the fields in the forms are completed. The findings in relation to these two attributes will be covered in turn.

Twenty studies used the first interpretation of the term in their assessment of surveillance systems [12,13,14,20,21,22,23,24,25,26,27,28,29,30,48,49,50,52,53,54]. They used retrospective analysis of routinely collected data [12,13,14,20,21,22,23,24,25,26,27,28,30,49,54] and retrospective analysis of routinely collected data combined with capture-recapture statistical methods [29,50,52,53], with only one study [48] opting to collect collected data prospectively. The proportion of cases captured by the different surveillance systems was compared with a variety of alternative sources of data, or estimated using the capture-recapture method, and ranged from 45% of cases in a TB surveillance system in Saudi Arabia [20] to 99.9% for a salmonellosis surveillance system in Sweden [52].

Most studies concluded that the completeness of the surveillance systems could be improved and made suggestions for how this could be achieved. This included switching to electronic reporting and training of personnel [27]; raising awareness among healthcare practitioners about the importance of reporting the conditions under surveillance [24]; incorporating data from other data collection systems (from primary care, hospitals, and pharmacies) [12,14,26,28,53]; making the notification process simpler [54].

The studies also identified characteristics that were associated with a higher or lower likelihood of cases being reported. An evaluation of the Swedish statutory surveillance system for communicable disease, which monitors salmonellosis and penicillin-resistant pneumococci, alongside other conditions, observed that it had higher sensitivity for recording diseases with a longer tradition of reporting (such as salmonellosis) [52]. An evaluation of TB notification in the Republic of Korea [21] noted that the type of medical institution (e.g., clinic or general hospital) and the nationality of the patient influenced the likelihood of cases being reported, with patients from general hospitals and those who were Korean nationals being more likely to be reported than those from clinics and foreign nationals respectively. The evaluation of a TB surveillance system in Italy [22] observed that under-notification was significantly higher in female patients and those with extra-pulmonary TB than in male patients and those with pulmonary TB, respectively. The surveillance of *Haemophilus* influenza was reported to have lower completeness following the introduction of routine immunisation in 1992, possibly leading to an over-estimation of the effectiveness of the vaccination programme [30].

Thirteen studies [13,15,17,20,22,26,31,32,33,34,35,36,54] investigated the second interpretation of the term “completeness” (the extent to which fields are completed in forms). All studies used retrospective analysis of routinely collected data to determine completeness. There was variation in how studies calculated the completeness of the fields in the case forms, with different approaches to classifying fields being used. For example, one study split fields into demographic and clinical variables [17] whilst another focused on a number of critical fields, for example, the site of the disease [31].

Completeness ranged from 34% to 90%, but the extent to which this was considered acceptably varied depending on the nature of the condition under surveillance, with a general view expressed that action was needed to be taken to improve rates. A concerning finding of the national TB surveillance system in Afghanistan was that the audit uncovered fields that were completed entirely incorrectly, whereby facilities reported examining sputum smears when they did not have a microscope [13].

#### 3.3.4. Concordance

Eight studies included this attribute [15,17,18,20,26,29,36,37], using retrospective analysis of routinely collected data [15,17,18,26,36,37] alongside one study that additionally combined this with capture-recapture statistical methods [29]. These studies established the accuracy of data within surveillance systems by exploring the level of agreement of data collected either at different organisational levels (e.g., facility and province) [18], from different sources (microbiologists or clinicians) [36,37] or in different formats (paper sources and electronic formats) [20,26]. In addition, data from surveillance systems were compared with mandatory notification systems [29]. Generally, the level of agreement between sources was found to be higher for patient demographic data (*k* > 0.80) than for clinical data [17,18,26,36,37]. Sprinson et al. [36] identified the importance of concordant data to enable programme planning and evaluations, particularly in areas such as policy development and programme advocacy. Podewils et al. [26] recommended unifying different systems (paper, electronic and laboratory) to reduce these discrepancies.

#### 3.3.5. Timeliness

This attribute was included in 18 studies [13,15,16,21,27,31,34,40,41,42,43,44,45,46,47,54,55,56], all using the same method (retrospective analysis of routinely collected data) but different reference periods to measure timeliness. These included the interval between the onset of symptoms and the notification to health authorities [40,43]; specimen collection to case reporting [27,42,54]; isolation at the primary laboratory to reporting to the surveillance body [47]; onset of symptoms to completion of case investigation [44]; reporting within one week of starting treatment [31]; starting treatment to notification [21]; notification within one or two incubation periods [40]; reporting delays regularly more than three months [13]. With the exception of one [16], all evaluations reported that timeliness of reporting could be improved, and some recommended that future research should explore the bottlenecks in the reporting process [41,42]. Suggestions made for improving the timeliness of reporting included transmitting laboratory reports electronically [56], reducing the laboratory electronic reporting period [55], and sending a sample of isolates to the national reference laboratory while the isolate is being analysed at the primary laboratory to reduce the delays incurred by waiting for the results of partial analysis [47].

#### 3.3.6. Positive Predictive Value (PPV)

This attribute was examined in five studies [13,14,24,35,36], using retrospective analysis of routinely collected data. There was wide variation, with the lowest value (1.1%) being reported for one variable in the TB surveillance system in California [36], whereas the TB Notification System of the United States Military scored 100% [14]. Stenhem et al. [35] observed that variation in the PPV (calculated by comparing the information in the first notification with the information in the study database) depended on the variable collected, Sprinson et al. [36] observed wider variations within their dataset when they compared the data on a sample of cases from a TB registry with medical records.

#### 3.3.7. Representativeness

Representativeness was included in only three evaluations [13,16,38], all using retrospective analysis of routinely collected data. An evaluation of a methicillin-resistant *Staphylococcus aureus* (MRSA) surveillance system in Japan using health insurance claims [38] observed that the surveillance system was accurately representing the ages of patients. Saeed et al. [13] observed that the HMIS in Afghanistan only included public health facilities with no data from the private sector (which it was attempting to cover). Furthermore, the system favoured rural areas with little attention to urban settings and secondary hospitals. Conversely, the evaluation of the Australian Gonococcal Surveillance Programme [16] observed that the reliance on molecular methods to diagnose infections had reduced the number of isolates available for AMR testing, especially in rural areas where molecular methods were increasingly used due to their greater cost-effectiveness.

#### 3.3.8. Acceptability

This attribute was examined in five studies through the use of surveys of users [16,55] and semi-structured interviews [13,15,54]. These found varying levels of acceptability related to awareness of the system and ease of use of the system. For example, the NTP in Afghanistan was found to have poor acceptability as health workers lacked awareness of the procedures to follow. Delayed reporting (and sometimes failure to report) indicated refusals to co-operate with the protocols [13]. The Australian Gonococcal Surveillance Programme had high acceptability for the reference laboratories that were contributors, even though some of the stakeholders identified the lack of feedback of surveillance data as an issue [16]. The acceptability of the salmonella surveillance system in London and South East England [55] was found to be related to the method used to follow up case reports, with telephone questionnaires being more successful than mailed ones.

An evaluation of a TB surveillance system in Brazil found that acceptability and timeliness were linked to each other when Brazil’s micro-regions were divided into two groups according to the relative performance of their systems [15].

#### 3.3.9. Flexibility

Two studies measured this attribute [13,16], using a survey of stakeholders [16] and interviews [13]. Samaan et al. observed that the reduced availability of isolates for testing (a result of the introduction of molecular-based methods to diagnose gonococcal infections) had challenged the flexibility of the Australian Gonococcal Surveillance Programme. It was trying to adapt to this situation by communicating with the public and private laboratories and asking them to forward any isolates they had available. Another challenge to the Australian system’s flexibility was the introduction of additional antimicrobials for the treatment of gonococcal infections. The system adapted through changes such as modifying its quality assurance process to include new resistance testing.

The two TB surveillance systems in Afghanistan evaluated by Saeed et al. [13] received different ratings, with the NTP being seen as flexible and able to accommodate changes. The HMIS, however, was seen as less flexible (for example, when there was a need for additional information or modes of operation).

#### 3.3.10. Simplicity

This was examined in three evaluations [13,16,54] that used questionnaire surveys [16] or interviews with stakeholders [13,54]. The evaluation of the *Campylobacter* Infection Surveillance Programme in Victoria (Australia) found it to be “cumbersome” to use for case referral and investigation [54]. Similarly, the Australian Gonococcal Surveillance Programme was observed by Samaan et al. [16] to have reduced the simplicity of the system as a result of duplication of data entry for the isolates received from the initial diagnostic laboratories. However, the survey of stakeholders undertaken as part of the same evaluation found that 88% of respondents felt that the system was sufficiently straightforward. Those who did not find it so cited the poorly defined terminology in reports as one of their concerns. Saeed et al. [13] also reported divergent findings in relation to the simplicity of two TB monitoring systems in Afghanistan (the HMIS and the NTP). The HMIS system was found to be simple as case definitions were followed and consistent forms were used. The NTP system, on the other hand, was found to be complex due to multiple complicated forms that need to be completed, the use of paper forms, a lack of integration with the HIMS system and difficulties in providing training for staff.

#### 3.3.11. Stability

Stability, defined as the ability to collect, manage, and provide data properly without failure and ability to be operational when needed [3], was included as an attribute in only one study [13], which investigated two TB surveillance programmes in Afghanistan using interviews. The authors reported that the HMIS was able to collect, manage and provide data from its facilities and produce monthly reports. The only problem reported that affected stability related to extraction of data from the system as a result of interruptions to the electricity supply. However, the authors also identified computer viruses as a potential threat. The other TB surveillance system in that evaluation was the NTP which was given a lower rating for stability due to being paper-based, possibly resulting in reporting delays.

## 4. Discussion

This review sought to examine the effectiveness of aspects of the performance of surveillance systems that are considered to be important for AMR monitoring. It found that most surveillance systems evaluated were underperforming in the key attributes related to both effectiveness and acceptability.

Lewis [57] described the overall objective of surveillance of AMR as the facilitation of control of AMR through informing the need to improve prescribing and infection control practices. The author identified three essential attributes for an AMR surveillance system for human health: timeliness, reliability, and representativeness. Timeliness was identified as important for AMR trends at the local level and to assist clinicians in the rational choice of antibiotic. However, the author acknowledged that prescribing decisions also needed to be supported by evidence on what constitutes unacceptable levels of resistance. Reliability (referring to the consistency in the laboratory data production process) was seen as important to assess trends over time and for benchmarking of resistance rates. The geographic, demographic and socioeconomic representativeness of the populations served by the laboratories where samples are generated was also seen to be important in order to be able to produce generalisable results. Our results indicate that for all three characteristics, the systems evaluated were underperforming.

There was wide variation in the nature and number of attributes examined in studies. Only two out of 43 reported all of the CDC attributes in their evaluations [45,52], with the majority focusing on only one or two attributes. Two characteristics, ease of use and awareness of the system, were associated with both greater acceptability and completeness (percentage of cases of condition reported). None of the other system characteristics were linked to benefits across more than one attribute. However, an examination of the attributes identifies other areas where the system’s performance on one attribute could influence others, even where this was not explicitly tested in the study. For example, Jansson et al. [42] observed that the timeliness of the Swedish statutory surveillance system was influenced by whether the reporting system was computerised or not, with computerised reporting resulting in shorter delays. The authors also point out in their discussion that computerised reporting would also be easier for clinicians to report cases. Hence, even though the authors did not examine simplicity as part of their evaluation, it is likely that their observations were relevant to this attribute.

There are two main reasons for caution when generalising the results of this review. Firstly, most evaluations (22/44) were of TB surveillance systems. This condition is at the forefront of the AMR challenge, as it is estimated that in 2017 there were 558,000 new cases of TB worldwide that were resistant to Rifampicin (the most effective first-line antibiotic), and among those, 82% had multidrug-resistant TB [58]. However, it is also a condition that is caused by a single pathogen, whereas the majority of infections (such as those affecting the upper and lower respiratory tracts, the urinary tract, wound, and bloodstream infections) are caused by a range of pathogens, and hence empirical treatment decisions require both the knowledge of the likely organisms and their likely susceptibilities to antibiotics [59].

Secondly, eight studies in the review were published between 1990 and 1999, and a further 14 were published between 2000 and 2009. Older evaluations may be of less relevance to certain aspects of current systems due to technological changes that have been implemented to the design and operation of surveillance systems which would make any older evaluations out of date and hence of limited relevance. Taking the UK as an example, one of the four British studies looked at the timeliness of reporting of *Salmonella* infections and the acceptability of the follow-up process [55]. The system, which used electronic reporting in 2010 in London and South East England, may not be relevant anymore given that the CoSurv software for recording laboratory isolates and case notifications that was examined in that study has since been replaced in England by the Second Generation Surveillance System (SGSS) Communicable Disease Module CDR [60]. The generalisability of a second study was also limited as the evaluation [25] was conducted between 1991–1993. The third study by Teo et al. [48], which investigated the Enhanced Tuberculosis Surveillance Scheme across England, Wales and Northern Ireland using a prospective rather than retrospective study design, was less prone to biases that may result from missing or erroneous entries. However, similar to the previous two, the age of the study (conducted between 2003 and 2005) limits its relevance to the current surveillance system in the UK. The current COVID-19 pandemic has demonstrated the impact that severe stresses can have on health systems. While laboratories remained largely functional during the pandemic, the surveillance reporting attached to these laboratories was deescalated due to capacity and other constraints for syndromic surveillance in particular [61].

There are also marked differences in the context of healthcare systems in the 18 countries where the surveillance systems were based, which would reduce the direct applicability of some of the findings to other settings. For example, the study by Olowokure et al. [30] to examine whether the introduction of the *Haemophilus* influenza type b conjugate vaccine in the UK in 1992 was associated with decreased effectiveness of the routine surveillance system, was confined to children under five living in one specific part of the UK (West Midlands).

### Strengths and Limitations of the Research

The main strength of this systematic review is that it analysed study findings at the level of the attributes of different systems derived from the most widely recognised guidelines for establishing the utility of the outputs of surveillance systems [3]. This had the advantage of permitting a more homogenous synthesis of findings across diverse evaluations in a manner that has not been reported previously and which enables the identification of the attributes that need to be more consistently assessed, reported and implemented.

A limitation of the study relates to the restriction of the database searches to studies published in English, so some relevant studies may have potentially been missed.

## 5. Conclusions and Recommendations

We conclude that most of the health surveillance systems relevant to AMR that we reviewed were under-performing in the main attributes that relate to effectiveness and acceptability. Given that all desirable attributes cannot be maximised simultaneously, policy-makers need to decide which are the priority features that they seek to include in health surveillance and monitoring systems. Ease of use and users’ awareness of surveillance systems have been shown to be linked to high levels of acceptability and higher levels of completeness of data collection and could be targeted as priority areas for improvement of existing systems.

In addition, we recommend consistent use of definitions of attributes of surveillance systems. This would ensure a more consistent approach to evaluations and facilitate the identification of the attributes that have the greatest impact on the utility of the data produced by these systems, and hence should be prioritised for monitoring, as well as those that are highly correlated with others, and could be given lower priority in evaluations.

## Figures and Tables

**Figure 1 antibiotics-10-00431-f001:**
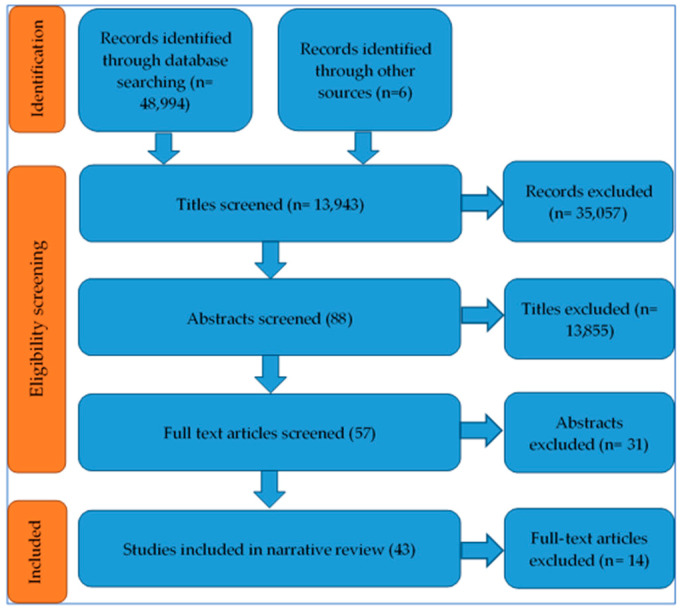
Preferred Reporting Items for Systematic Reviews and Meta-Analyses (PRISMA) diagram of studies included at each stage of the screening process.

**Table 1 antibiotics-10-00431-t001:** Number of studies included in the review by study design.

Study Design and Analysis Approaches	Number of Studies
Observational: Retrospective analysis of routinely collected data	30
Mixed methods: Retrospective analysis of routinely collected data combined with semi-structured interviews	5
Observational: Retrospective analysis of routinely collected data combined with capture-recapture statistical methods	5
Observational: Retrospective analysis of routinely collected data combined with a questionnaire survey	2
Observational: Prospective analysis of routinely collected data	1
Total	43

**Table 2 antibiotics-10-00431-t002:** Country where evaluation took place.

Country of Study	Number of Studies
Afghanistan	1
Australia	4
Brazil	3
France	3
Germany	1
Ireland	1
Italy	1
Japan	1
Netherlands	2
Republic of Korea	1
Romania	1
Saudi Arabia	1
South Africa	3
Spain	2
Sweden	3
Taiwan	1
UK	4
USA	10
Total	43

**Table 3 antibiotics-10-00431-t003:** Attributes identified and their definitions.

Attribute Name.	Description of Attribute as Used by Evaluators
Acceptability	Awareness of, and adherence to, the surveillance system protocol by staff.
Completeness (also described as sensitivity, coverage, validity)	Either: The proportion of cases reported by the system (established by looking at other systems or by estimating using the capture-recapture method); also known as sensitivity or coverage.
	Or: Extent (or proportion) of the fields that are completed in the forms. In some studies, critical categories to be completed were identified; also known as validity.
Concordance (also known as reliability or consistency)	The level of agreement between the different systems on the data collected for each case.
Flexibility	The degree to which a system can adapt to changing information needs or operating conditions with little additional time, personnel, or allocated funds (CDC Definition) [3].
Positive Predictive Value (also known as Predictive Value Positive) PPV	The proportion of reported cases that actually have the health-related event under surveillance (CDC Definition) [3].
Representativeness	Geographic or population coverage of system.
Simplicity	Features that make a system easy to use (including the method of notification).
Specificity	Correctly identifying patients who are free of the condition.
Stability	Ability to collect, manage, and provide data properly without failure and ability to be operational when needed [3].
Timeliness	Period between different time points in the notification process.
Usefulness	Ability of a system to provide information that can be (or is) acted on; also known as efficacy.

**Table 4 antibiotics-10-00431-t004:** Antimicrobial resistance (AMR)-related health conditions and microorganisms monitored in surveillance systems. TB: tuberculosis; MRSA: methicillin resistance *Staphylococcus aureus*.

Health Condition/Microorganism	Number of Included Evaluations
TB (Pulmonary or extra-pulmonary)	22
*Salmonella*/salmonellosis	8
Infections with penicillin-resistant pneumococci	2
MRSA	3
*Neisseria gonorrhoeae*/Gonococcal infections	2
Shiga-toxin producing or enterhaemorrhagic *Escherichia coli*	2
Shigellosis	1
TB in HIV patients	4
*Campylobacter*	1
*Haemophilus influenzae*	1

## Data Availability

Data is contained within the article.

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
