# Peer review of "Quality and Utility of Information Captured by Surveillance Systems Relevant to Antimicrobial Resistance (AMR): A Systematic Review"

_antibiotics, 2021, doi:10.3390/antibiotics10040431_

Round 1

Reviewer 1 Report

The manuscript is a systematic review of the quality and utility of information captured by surveillance systems relevant to antimicrobial resistance (AMR). Overall, the manuscript is clearly written, extensively detailed, and is likely to be of interest to a diverse readership, including those interested in antibiotic resistance mechanisms. Although I do not have any strong criticism for the manuscript, I find lots of formatting errors (spacing problem) in the manuscripts which need to be fixed before further consideration.

Minor issues:

  1. Including a paragraph about the reasons behind increasing antimicrobial resistance could help the reader.
  2. Line 75: total add up to 49000.

Apart from these authors need to be congratulated for their commendable effort in putting together this systematic review article.

Reviewer 2 Report

-This is an important topic in combating AMR.

  • The authors did a great job in writing the manuscript esp in tabulating the attributes for definition/description as readers may not comprehend such terms.
  • Structure of the manuscript should be revised- Materials and Methods should be before Results
  • Strengths and limitations should be part of the Discussion
  • Part of the limitation should also include cost, resources needed to train personnel, etc. 
  • What about conclusion?
